# Characterization of Single-Nucleus Electrical Properties by Microfluidic Constriction Channel

**DOI:** 10.3390/mi10110740

**Published:** 2019-10-31

**Authors:** Hongyan Liang, Yi Zhang, Deyong Chen, Huiwen Tan, Yu Zheng, Junbo Wang, Jian Chen

**Affiliations:** 1State Key Laboratory of Transducer Technology, Aerospace Information Research Institute, Chinese Academy of Sciences, Beijing 100094, China; lianghongyan17@mails.ucas.edu.cn (H.L.); zhangyi161@mails.ucas.edu.cn (Y.Z.); dychen@mail.ie.ac.cn (D.C.); tanhuiwen18@mails.ucas.edu.cn (H.T.); 2School of Electronic, Electrical and Communication Engineering, University of Chinese Academy of Sciences, Beijing 101408, China; 3Shandong University, Jinan 250100, China; 201800301017@mail.sdu.edu.cn

**Keywords:** microfluidics, single-nucleus analysis, constriction channel, electrical properties, nuclear envelope

## Abstract

As key bioelectrical markers, equivalent capacitance (C_ne_, i.e., capacitance per unit area) and resistance (R_ne_, i.e., resistivity multiply thickness) of nuclear envelopes have emerged as promising electrical indicators, which cannot be effectively measured by conventional approaches. In this study, single nuclei were isolated from whole cells and trapped at the entrances of microfluidic constriction channels, and then corresponding impedance profiles were sampled and translated into single-nucleus C_ne_ and R_ne_ based on a home-developed equivalent electrical model. C_ne_ and R_ne_ of A549 nuclei were first quantified as 3.43 ± 1.81 μF/cm^2^ and 2.03 ± 1.40 Ω·cm^2^ (N_n_ = 35), which were shown not to be affected by variations of key parameters in nuclear isolation and measurement. The developed approach in this study was also used to measure a second type of nuclei, producing C_ne_ and R_ne_ of 3.75 ± 3.17 μF/cm^2^ and 1.01 ± 0.70 Ω·cm^2^ for SW620 (N_n_ = 17). This study may provide a new perspective in single-cell electrical characterization, enabling cell type classification and cell status evaluation based on bioelectrical markers of nuclei.

## 1. Introduction

Nuclear envelope defines a bilayer membrane that encloses the genome from the rest of the cell and regulates the movements of molecules across the nuclear-cytoplasmic boundary. As key bioelectrical markers, equivalent capacitance (C_ne_, i.e., capacitance per unit area) and resistance (R_ne_, i.e., resistivity multiply thickness) of the nuclear envelope have emerged as promising electrical indicators related to salivary gland cells with changes in developments [1,2,3,4,5,6], oocytes isolated from different species [7,8,9,10,11,12,13], normal and malignant white blood cells [14,15,16,17,18,19,20], yeasts with changes in extracellular solutions [21,22], epithelia cell lines from multiple sources [23,24,25,26,27,28,29,30,31,32,33,34,35,36] and differentiation of stem cells [37]. A summary of previously reported electrical parameters of single nuclei could be found in Appendix A.

Patch clamping was initially adopted to characterize electrical properties of nuclear envelopes, where two glass pipettes were deployed within the nuclear and cytoplasmic domains, respectively, enabling the quantitation of C_ne_ and R_ne_ [1,2,3,4,5,6,7,8,9,23,38,39,40]. Based on this approach, C_ne_ and R_ne_ of single nuclei were characterized as 412 ± 62 μF/cm^2^ and 1.5 ± 0.4 Ω·cm^2^ [1], ~100 μF/cm^2^ and 3.9 ± 1.4 Ω·cm^2^ [2], 0.72 ± 0.09 Ω·cm^2^ [4] and 2 Ω·cm^2^ [6] of salivary gland cells of *Drosophila flavorepleta*. Although powerful, patch clamping was mostly used to measure ultra large cells with diameters of nearly 100 μm and it is full of challenges to accurately penetrate nuclear domains of conventional eukaryotic cells with diameters around 10 μm.

In a second approach, two electrodes were inserted into cell suspensions and the corresponding impedance data were interpreted into both electrical parameters of cell membranes and nucleus [15,17,18,21,24,25,28,29,31,34,36,41,42,43,44,45]. Based on this approach, C_ne_ and R_ne_ of nuclei derived from mouse spleen lymphocytes, human Jurkat cells and rat H9C2 cells were quantified as 0.62 μF/cm^2^ and 0.07 Ω·cm^2^ [15], 1.19 ± 0.14 μF/cm^2^ and 0.21 ± 0.02 Ω·cm^2^ [36], 0.22 ± 0.05 μF/cm^2^ and 7.25 ± 0.68 Ω·cm^2^ [34], respectively. However, this is an approach based on cell populations rather than single cells where potential concerns of interactions among neighboring cells during measurements cannot be properly addressed.

In order to address this issue, electrorotation was adopted for the characterization of single-nucleus electrical properties where single cells were forced to rotate by four electrodes and the rotating speeds as a function of the applied electrical signals were used to estimate electrical properties of nuclear envelopes [14,16,19,25,46]. However, in this approach, a double shell electrical model representing cell and nuclear membranes was used. Thus, capacitive properties of cell membranes may mask electrical properties of nuclear membranes at the low-frequency domain, while at the high-frequency domain, both cell membranes and nuclear membranes are short circuited.

To address this issue, in this study, cell lysis and nucleus isolation were first conducted and the obtained single nuclei were trapped at the entrance of the constriction channel for electrical property characterization (see Figure 1). In operation, trypsin-EDTA, the lysis buffer and IGEPAL^@^ CA-630 were used to acquire nuclei of cells (see Figure 1a). Then individual nucleus was trapped at the entrance of the constriction channel with corresponding impedance values measured by a lock-in amplifier (see Figure 1b). Based on an equivalent electrical model where nuclear electrical components were represented as C_ne_, R_ne_ and R_np_ (equivalent resistance of nucleoplasm), raw impedance data were translated into C_ne_ and R_ne_ (see Figure 1c).

In comparison to the aforementioned population approach for nuclear electrical property characterization, this constriction channel based approach enables the electrical property characterization at single nucleus level. In comparison to electrorotation, in this study, side effects of membrane capacitance on the estimation of nuclear electrical properties are addressed. In comparison to patch clamping, this device can easily trap single nuclei at the entrance of the constriction channel without the requirements of accurate manipulations of pipette tips.

## 2. Materials and Methods

### 2.1. Materials

All cell lines were purchased from China Infrastructure of Cell Line Resources. All reagents for cell culture (e.g., culture medium, fetal bovine serum and trypsin) were purchased from Life Technologies Corporation (Van Allen Way Carlsbad, CA, USA). All reagents for cell treatments (e.g., isotonic lysis buffer, dithiothreitol, protease inhibitor cocktail and IGEPAL^@^ CA-630) were purchased from Sigma Aldrich Corporation (St. Louis, MO, USA). The materials required for device fabrication included SU-8 photoresist (MicroChem Corporation, Newton, MA, USA) and 184 silicone elastomer (Dow Corning Corporation, Midland, MI, USA).

### 2.2. Nucleus Preparation

A549 and SW620 cancer cell lines were cultured in RPMI 1640 supplemented with 10% fetal bovine serum, 1% penicillin and streptomycin, under the conditions of 37 °C and 5% CO_2_. For nucleus preparation, it was adopted from a previous study [47] where the cultured cancer cells were trypsinized using 1x trypsin-EDTA from culture flasks, which were then incubated with an isotonic lysis buffer supplemented with 1 mM dithiothreitol and protease inhibitor cocktail on ice for 15 min. After that IGEPAL^@^ CA-630 was added into the cell suspension, followed by a vortex for 10 s, and then a centrifugation of 400× *g* for 5 min. In the end, isolated nuclei were resuspended in 1x phosphate buffer saline containing 1% bovine serum albumin (see Figure 1a). Note that in nuclear isolation, variations of IGEPAL^@^ CA-630 were used for comparison where 0.01%, 0.02% and 0.03% of IGEPAL^@^ CA-630 were used for A549 cells while 0.01% of IGEPAL^@^ CA-630 was used for SW620 cells.

### 2.3. Device Fabrication

The device mainly consists of constriction channels (cross-sectional dimensions of 7 μm × 8 μm for A549 nuclei or 5 μm × 5 μm for SW620 nuclei) in polydimethylsiloxane (PDMS) elastomers, which were replicated from SU-8 mold masters using conventional soft lithography. Briefly, SU-8 5 was spin-coated, exposed without development to form the layer of the constriction channel with a height of 7 μm or 5 μm. Then, SU-8 25 was spin coated, exposed with alignment and developed to form the nucleus-loading channel with a height of 25 μm. After fabricating SU-8 mold masters, PDMS precursor and curing agents (ratio 10:1 by weight) were thoroughly mixed, degassed and poured onto the SU-8 channel masters for crosslinking (4 hours at 80 °C). Fully cured PDMS channels were then peeled away, punched with through holes as inlets and outlets, and bonded to glass slides after plasma treatment.

### 2.4. Device Operation

In experiments, the microfabricated channels were first filled with 1× phosphate buffer saline containing 1% bovine serum albumin and then loaded with nuclei at a concentration of 1 × 10^6^ nuclei/mL. A pressure controller (Pace 5000, Druck, Billerica, MA, USA) was used to generate negative pressures to trap single nuclei at the entrance of the constriction channels (0.2, 0.5 and 1.0 kPa for A549 nuclei at 7 μm × 8 μm constriction channels; 0.5, 1.0 and 2.0 kPa for SW620 nuclei at 5 μm × 5 μm constriction channels). Then, impedance profiles from 1 kHz to 250 kHz (excitation voltage: 200 mV) were recorded by a lock-in amplifier (7270, Signal Recovery, Oak Ridge, TN, USA). Meanwhile, an inverted microscope (IX83, Olympus, Tokyo, Japan) was used to capture the images of trapped single nuclei. After characterization, high negative pressures were used to aspirate the nuclei through the constriction channels and then the device was ready for the next measurements (see Figure 1b).

### 2.5. Data Processing

To interpret the measured impedance data, an electrical model was proposed (see Figure 1c). The electrical model of the constriction channel was represented by an equivalent resistor (R_c_) and a capacitor (C_c_) in parallel. Nuclear electrical components were represented as a capacitor (C_ne_) in parallel with a resistor (R_ne_) for the portion of the nuclear envelope in series with a resistor (R_np_) for the nucleoplasm portion. Furthermore, an equivalent leakage resistor (R_l_) was defined to estimate sealing status between the aspirated membrane portion of the nucleus under the measurement and inner walls of the constriction channel.

Impedance profiles without nucleus trapping were first fitted with the equivalent electrical components of the constriction channel to obtain values of equivalent resistor (R_c_) and capacitor (C_c_). Then impedance profiles with nucleus trapping were fitted with the aforementioned electrical model, based on the nonlinear least-square principle, where a loop function was used to enumerate key parameters of C_ne_, R_ne_, R_np_ and R_l_. Note that in the step of curve fitting of impedance profiles with nucleus trapping, R_c_ and C_c_ were treated as known variables without looping and thus the potential concern of parasitic capacitors of constriction channels on the extraction of electrical parameters of nuclear envelopes can be properly addressed.

### 2.6. Statistics

The measurements of multiple samples were conducted with results expressed by averages and standard deviations. The student’s *t*-test was used, where the values of *p* < 0.001 (*) were considered as statistically significant.

## 3. Results and Discussion

Microfluidics refer to the manipulation of microscale fluids in microfabricated channels [48]. Due to dimensional comparison with cells, microfluidics has functioned as an enabling tool for single-cell isolation [49,50,51,52], and then impedance measurements [53]. Recently, single cells were trapped by a microfluidic device with corresponding impedance values measured, enabling the quantification of single-nucleus electrical properties [37]. However, in this approach, the electrical parameters of cell membranes can have side effects on the characterization of nuclear electrical properties. In this study, nuclei were isolated from cells and then trapped at the entrance of the constriction channel for electrical property characterization and thus the potential concern on the cell membranes can be properly addressed.

Figure 2 shows the microscopic images of isolated single nuclei stained with trypan blue, where stained blue dots and non-stained counterparts in the images represent isolated nuclei and intact cells, respectively. Under the conditions of 0.01%, 0.02% and 0.03% IGEPAL^@^ CA-630 for the treatments of A549 cells, it was observed that the increase of the percentage of IGEPAL^@^ CA-630 enhanced the ratio of cell lysis and percentage of nuclear isolation (see Figure 2a–c). As to SW620 cells, 0.01% IGEPAL^@^ CA-630 can produce comparable results with A549 cells under the treatment of 0.02% IGEPAL^@^ CA-630 (see Figure 2d).

Figure 3 shows measured impedance values with curve fitting and correspond microscopic images for trapped single nuclei at the entrance of the constriction channels. As shown in Figure 3a, for A549 nuclei, under the conditions of 0.01% IGEPAL^@^ CA-630 and a constriction-channel of 7 μm × 8 μm, with the increase of aspiration pressure (P_a_), impedance amplitude and phase values were noticed to increase and decrease, respectively. Meanwhile, the length of the nucleus aspirated into constriction channel extended.

Curve fitting (fit-base line) of measured impedance data without trapped single nuclei at the entrance of the constriction channel was conducted where C_c_ and R_c_ were quantified as 0.20 pF and 1.19 MΩ, respectively. Curve fitting (fit-0.2 kPa line, fit-0.5 kPa line and fit-1.0 kPa line) of measured impedance data with trapped single nuclei at the entrance of the constriction channel was conducted where C_ne_, R_ne_, R_np_ and R_l_ were quantified as 1.34 μF/cm^2^, 4.48 Ω·cm^2^, 0.50 MΩ and 0.45 MΩ (P_a_ = 0.2 kPa), 1.88 μF/cm^2^, 1.85 Ω·cm^2^, 0.60 MΩ and 0.80 MΩ (P_a_ = 0.5 kPa), 1.88 μF/cm^2^, 3.72 Ω·cm^2^, 0.60 MΩ and 0.95 MΩ (P_a_ = 1.0 kPa; see Figure 3a)

As shown in Figure 3b, a similar trend was obtained for A549 nuclei under the conditions of 0.02% IGEPAL^@^ CA-630 and the constriction-channel of 7 μm × 8 μm. As to A549 nuclei under the condition of 0.03% IGEPAL^@^ CA-630, no significant differences of impedance with and without the trapping of single nucleus at the entrance of the constriction channel was noticed, indicating that the use of high concentration of IGEPAL^@^ CA-630 can damage nuclear envelopes, which then cannot effectively block the electric lines in impedance measurements (see Figure 3c).

As shown in Figure 3d, a similar trend was obtained for SW620 nucleus under the conditions 0.01% IGEPAL^@^ CA-630 and the constriction-channel of 5 μm × 5 μm. Curve fitting of measured impedance data for single nucleus was conducted where C_ne_, R_ne_, R_np_ and R_l_ were quantified to be 3.33 μF/cm^2^, 1.83 Ω·cm^2^, 0.20 MΩ and 0.65 MΩ (P_a_ = 0.5 kPa), 4.00 μF/cm^2^, 0.62 Ω·cm^2^, 0.20 MΩ and 1.75 MΩ (P_a_ = 1.0 kPa), 3.50 μF/cm^2^, 2.64 Ω·cm^2^, 0.40 MΩ and 1.80 MΩ (P_a_ = 2.0 kPa).

Figure 4 shows quantified single-nucleus electrical parameters of C_ne_, R_ne_ and R_np_ as well as R_l_. Under the conditions of 0.01% IGEPAL^@^ CA-630 and the constriction-channel of 7 μm × 8 μm, C_ne_, R_ne_, R_np_ and R_l_ of the A549 nuclei were determined to be 4.02 ± 1.91 μF/cm^2^, 1.67 ± 1.37 Ω·cm^2^, 0.41 ± 0.21 MΩ and 0.43 ± 0.30 MΩ (N_n_ = 16, P_a_ = 0.2 kPa), 3.55 ± 1.66 μF/cm^2^, 2.20 ± 1.52 Ω·cm^2^, 0.43 ± 0.17 MΩ and 0.81 ± 0.57 MΩ (N_n_ = 16, P_a_ = 0.5 kPa), 3.94 ± 2.16 μF/cm^2^, 2.82 ± 1.43 Ω·cm^2^, 0.47 ± 0.17 MΩ and 1.49 ± 1.34 MΩ (N_n_ = 16, P_a_ = 1.0 kPa; see Figure 4a and Table 1).

Under an arbitrary aspiration pressure, the obtained C_ne_ and R_ne_ were around 1 μF/cm^2^ and 1 Ω·cm^2^, which were consistent with previous studies [17,18,26,34,36,41]. With the increase of the aspiration pressure, R_l_ was observed to increase, suggesting the improvement in the sealing status between nuclear envelopes and inner walls of the constriction channel. Though the values of R_ne_ were noticed to increase when the aspiration pressure was increased from 0.2 kPa to 0.5 kPa and then 1.0 kPa, no differences with statistical significances for both C_ne_ and R_ne_ were located neighboring pressure values, indicating that the variations of aspiration pressure have no significantly side effects on the measurements of single-nucleus electrical properties.

Figure 4b shows quantified single-nucleus electrical parameters of C_ne_, R_ne_ and R_np_ as well as R_l_ under the conditions of 0.02% IGEPAL^@^ CA-630 and the constriction-channel of 7 μm × 8 μm. C_ne_ and R_ne_ of the A549 nuclei derived from 0.01% vs. 0.02% IGEPAL^@^ CA-630 were compared as follows: 4.02 ± 1.91 μF/cm^2^ vs. 2.97 ± 1.32 μF/cm^2^ and 1.67 ± 1.37 Ω·cm^2^ vs. 1.45 ± 1.07 Ω·cm^2^ (P_a_ = 0.2 kPa), 3.55 ± 1.66 μF/cm^2^ vs. 2.90 ± 1.47 μF/cm^2^ and 2.20 ± 1.52 Ω·cm^2^ vs. 1.76 ± 1.14 Ω·cm^2^ (P_a_ = 0.5 kPa) and 3.94 ± 2.16 μF/cm^2^ vs. 3.38 ± 1.96 μF/cm^2^ and 2.82 ± 1.43 Ω·cm^2^ vs. 2.38 ± 1.37 Ω·cm^2^ (P_a_ = 1.0 kPa; see Figure 4b and Table 1). Though the values of C_ne_ and R_ne_ were observed to decrease when the percentage of IGEPAL^@^ CA-630 was increased from 0.01% to 0.02%, no differences with statistical significances were located between these two groups, indicating that the use of IGEPAL^@^ CA-630 for cell lysis at low concentrations has no significantly side effects on the measurements of single-nucleus electrical properties.

Figure 4c shows quantified single-nucleus electrical parameters of C_ne_, R_ne_ and R_np_ as well as R_l_ under the conditions of 0.01% IGEPAL^@^ CA-630 and the constriction-channel of 5 μm × 5 μm for SW620 nuclei. Under an arbitrary aspiration pressure, the obtained C_ne_ and R_ne_ were around 1 μF/cm^2^ and 1 Ω·cm^2^, which were consistent with previous studies [17,18,26,34,36,41]. Again, no differences with statistical significances were located among values of C_ne_ and R_ne_ under three different aspiration pressures, confirming that the variations of aspiration pressure have no significantly side effects on electrical properties of nuclear envelopes. Compared with the values of C_ne_ and R_ne_ obtained from A549 nuclei, the obtained values of SW620 nuclei were lower in both C_ne_ (4.00 ± 3.03 μF/cm^2^ vs. 3.75 ± 3.17 μF/cm^2^) and R_ne_ (1.69 ± 1.29 Ω·cm^2^ vs. 1.01 ± 0.70 Ω·cm^2^). Furthermore, significant differences with statistics were only located in R_ne_, suggesting that nuclei of A549 and SW620 are different in equivalent resistance of nuclear envelopes while they have comparable capacitive properties.

## 4. Conclusions and Future Developments

This study demonstrated the feasibility of the microfluidic device by quantifying electrical properties of A549 nuclei, and reporting electrical properties of single nuclei independent from key parameters in nuclear isolation and measurement. Then this device was also used to characterize electrical properties of SW620 nuclei, which indicated that this approach could be used to test multiple nucleus types. 

Future technical developments may aspirate single nuclei rapidly through the microfluidic constriction channel, enabling high-throughput quantification of single-nucleus electrical properties. Then the approach can be further expanded to high-throughput quantify multiple electrical parameters of cell membrane and nuclear portions, enabling cell type classification and cell status evaluation in a label-free manner.

## Figures and Tables

**Figure 1 micromachines-10-00740-f001:**
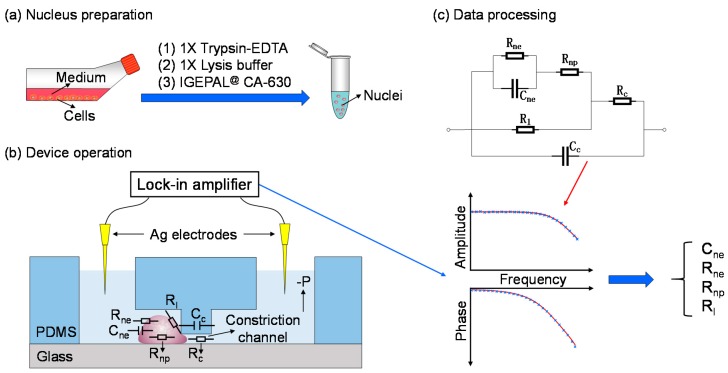
Working flowchart for characterizing single-nucleus electrical properties (e.g., C_ne_ and R_ne_, two key parameters of nuclear envelope) based on microfluidic constriction channel. Key steps include nucleus preparation (**a**), device operation (**b**) and data processing (**c**). In operation, trypsin-EDTA (Ethylenediaminetetraacetic acid), lysis buffer and IGEPAL^@^ CA-630 were used to acquire nuclei of cells. Then single nuclei were trapped at the entrance of the constriction channel with corresponding impedance values measured by a lock-in amplifier. Based on an equivalent electrical model where nuclear electrical components were represented as C_ne_ in parallel with R_ne_, and then in series with R_np_, raw impedance data were translated into C_ne_ and R_ne_.

**Figure 2 micromachines-10-00740-f002:**
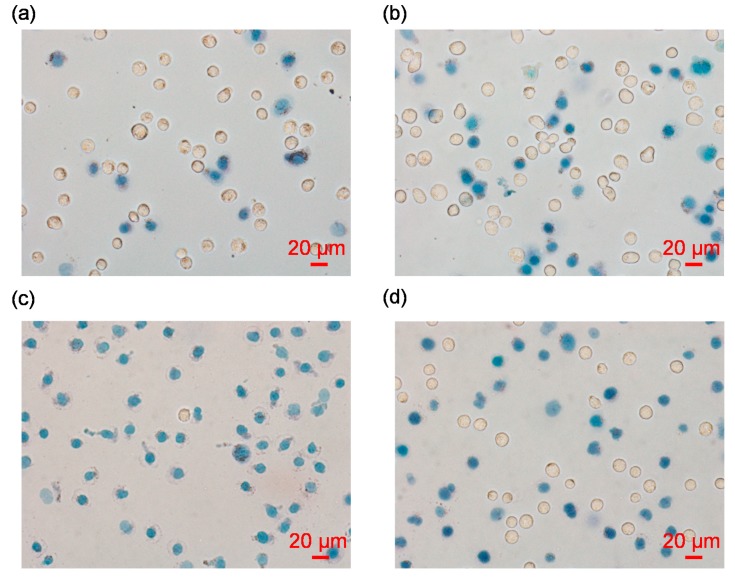
Microscopic images of isolated single nuclei stained with trypan blue, under the conditions of 0.01% IGEPAL^@^ CA-630 for A549 cells (**a**), 0.02% IGEPAL^@^ CA-630 for A549 cells (**b**), 0.03% IGEPAL^@^ CA-630 for A549 cells (**c**) and 0.01% IGEPAL^@^ CA-630 for SW620 cells (**d**). Note that stained blue dots and non-stained counterparts in the images represent isolated nuclei and intact cells, respectively.

**Figure 3 micromachines-10-00740-f003:**
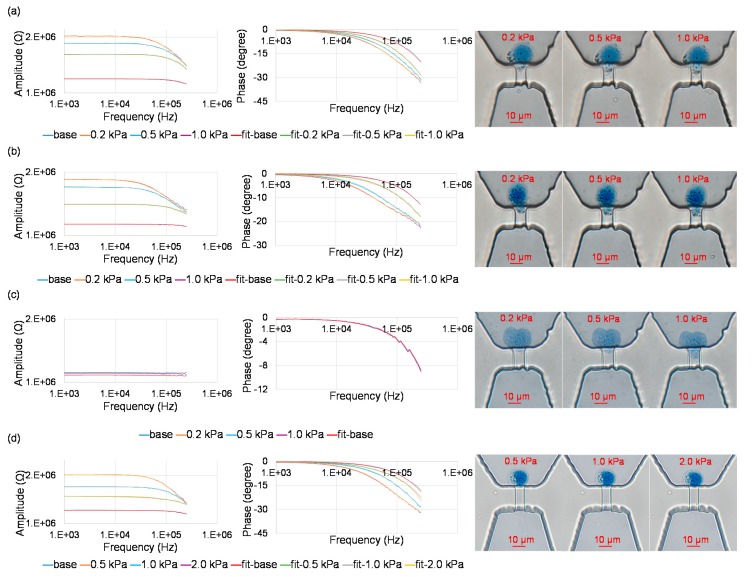
Measured impedance values with curve fitting and corresponding microscopic images for trapped single nuclei at the entrances of the constriction channels, under the conditions of IGEPAL^@^ CA-630: 0.01%, constriction channel: 7 μm × 8 μm, aspiration pressure: 0.2 kPa, 0.5 kPa and 1.0 kPa, nucleus type: A549 (**a**), IGEPAL^@^ CA-630: 0.02%, constriction channel: 7 μm × 8 μm, aspiration pressure: 0.2 kPa, 0.5 kPa and 1.0 kPa, nucleus type: A549 (**b**), IGEPAL^@^ CA-630: 0.03%, constriction channel: 7 μm × 8 μm, aspiration pressure: 0.2 kPa, 0.5 kPa and 1.0 kPa, nucleus type: A549 (**c**), IGEPAL^@^ CA-630: 0.01%, constriction channel: 5 μm × 5 μm, aspiration pressure: 0.5 kPa, 1.0 kPa and 2.0 kPa, nucleus type: SW620 (**d**).

**Figure 4 micromachines-10-00740-f004:**
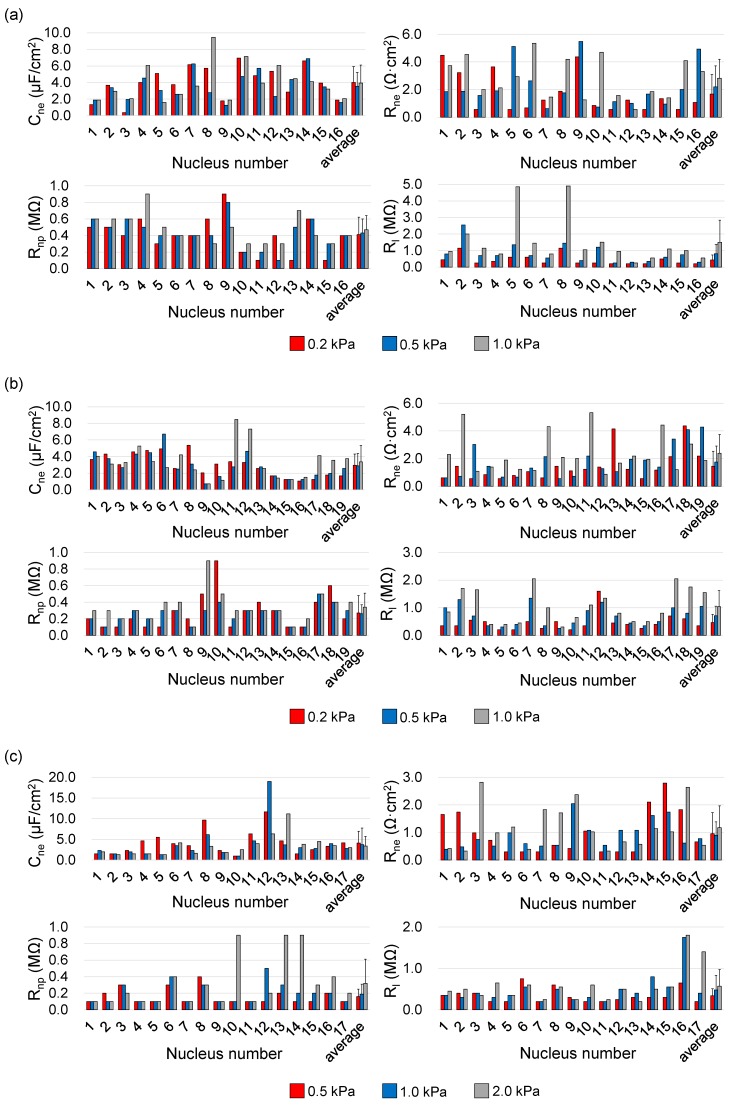
Quantified single-nucleus electrical parameters of C_ne_, R_ne_ and R_np_ as well as R_l_ under the conditions of IGEPAL^@^ CA-630: 0.01%, constriction channel: 7 μm × 8 μm, aspiration pressure: 0.2 kPa, 0.5 kPa and 1.0 kPa, nucleus type: A549, nucleus number: 16 (**a**), IGEPAL^@^ CA-630: 0.02%, constriction channel: 7 μm × 8 μm, aspiration pressure: 0.2 kPa, 0.5 kPa and 1.0 kPa, nucleus type: A549, nucleus number: 19 (**b**), IGEPAL^@^ CA-630: 0.01%, constriction channel: 5 μm × 5 μm, aspiration pressure: 0.5 kPa, 1.0 kPa and 2.0 kPa, nucleus type: SW620, nucleus number: 17 (**c**).

**Table 1 micromachines-10-00740-t001:** A summary of quantified single-nucleus electrical parameters (e.g., C_ne_, R_ne_ and R_np_) using constriction channels under a variety of operation conditions.

Nucleus Type	Constriction Channel Dimensions	IGEPAL^@^ CA-630	Aspiration Pressure (kPa)	C_ne_ (μF/cm^2^)	R_ne_ (Ω·cm^2^)	R_np_ (MΩ)	Nucleus Number
A549	7 μm × 8 μm	0.01%	0.2	4.02 ± 1.91	1.67 ± 1.37	0.41 ± 0.21	*N* = 16
			0.5	3.55 ± 1.66	2.20 ± 1.52	0.43 ± 0.17	
			1.0	3.94 ± 2.16	2.82 ± 1.43	0.47 ± 0.17	
		0.02%	0.2	2.97 ± 1.32	1.45 ± 1.07	0.27 ± 0.21	*N* = 19
			0.5	2.90 ± 1.47	1.76 ± 1.14	0.26 ± 0.11	
			1.0	3.38 ± 1.96	2.38 ± 1.37	0.34 ± 0.17	
SW620	5 μm × 5 μm	0.01%	0.5	4.13 ± 2.82	0.96 ± 0.76	0.16 ± 0.09	*N* = 17
			1.0	3.74 ± 4.03	0.90 ± 0.48	0.19 ± 0.12	
			2.0	3.38 ± 2.37	1.18 ± 0.79	0.32 ± 0.29	

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
