# Peer review of "Characterization of Single-Nucleus Electrical Properties by Microfluidic Constriction Channel"

_micromachines, 2019, doi:10.3390/mi10110740_

Round 1

Reviewer 1 Report

The authors have developed a method of single-cell nucleus electrical characterization wherein the issues of the influence of neighboring cells and capacitance from cell membranes is addressed. The authors have developed a device with constriction chambers using PDMS, wherein a single nuclei is constricted in a channel using a pressure controller and characterized using a lock-in amplifier. With their developed system, the authors have characterized the nucleus electrical characteristics of the two cancer cell types, A549 and SW620, and have acquired data that is consistent with previous studies, thus demonstrating the efficacy of their presented work.

Comments:

It would be beneficial for the authors to revise figures 3 and 4 to improve the readability of their data. Some grammatical errors are found. In Intro line 33, the capacitance is claimed to be a promising indicator for each variation. This sentence needs to be more specific. I would suggest adding specific variations for each cell type the authors are referring to that the capacitance can indicate, rather than to just say unspecific “variations”. The claims in line 57-59 need more specifics to be well understood by the readership. What is the equivalent circuitry that electrorotation is studying? Any capacitive system can be said to be short-circuited in extremely high frequency but that doesn’t mean that the measurement is useless in the high-frequency domain. In electrophysiology, it is common to distinguish the buffer/solution used for intracellular and extracellular measurements. For example, when performing a whole-cell patch-clamp, the buffer in the pipette would be the intracellular buffer while the bath solution would be the extracellular buffer to reflect their physiological conditions. And, of course, the ion compositions/concentrations vary between the two buffers. In the presented work, the nuclei are being extracted for the measurements. Thus, it would make sense to use the intracellular buffer to maintain their physiological condition in the fluidic channel. However, the work uses PBS. Unless the cell types used in this work have intracellular compositions that closely matches that of PBS, the result presented here might be artificially influenced. I would suggest one of two options. Either state that PBS can represent their physiological condition (by checking their ion compositions), or perform an experiment to show that changing the ionic composition (in case PBS and their intracellular buffer are very different) does not influence the capacitance measurement drastically to validate the presented data. There is a good possibility that the latter is true, given that osmolarity is usually conserved between the intracellular buffer and PBS, and I assume that osmolarity may be the biggest influencer to the capacitance measurements. There must be a purpose of measuring this capacitance and resistance. The author gives a vague claim in the intro saying that these values reflect their variations. What is the significance of this work? What does this data mean in terms of cell physiology for the given cell types? Is there a biological process or pharmaceutical modifications in which these electrical characteristics may vary that the authors want to apply this technology? If so, provide some examples. In the same context as comment 6, why were A549 and SW620 chosen? Do they carry any significance for this type of measurements?

Reviewer 2 Report

The manuscript entitled “Characterization of Single-Nucleus Electrical Properties by Microfluidic Constriction Channel“ by Liang et al. presents the a microfluidic platform for measuring electrical properties of single nuclei. While the authors presented some interesting results, the comparison with existing methods in literature is missing. Also, the precision and throughput seem to be limited. It is recommended that this paper should be significantly revised. Detailed comments are listed below:

The biological and clinical implications of single-nucleus electrical properties are not well-articulated. It is recommended that the authors can better explain this in the Introduction section. There are many existing works using similar geometry to isolate cellular objects in microfluidics. It is recommended that the authors should provide sufficient background introduction and include relevant references in single-cell isolation. It is recommended that the authors can better explain how to exclude parasitic capacitance/impedance of the microfluidic channels from nucleus electrical impedance with some control experiments. It is recommended that the authors can compare the measured single-nucleus electrical properties with previously reported values in literature to support the accuracy of presented method. In the data presented in Table 1, large heterogeneity was observed. It is recommended that the authors can provide some evidence to indicate whether the heterogeneity was the intrinsic property of cellular heterogeneity or was caused by artifact in measurement. In the conclusion, the authors argue the potential for high-throughput quantification of single-nucleus electrical properties. However, in this work, the authors only presented tens of data points. It is not convincing the presented technology has the potential for high-throughput measurement. The font size in Fig. 3a-d and Fig. 4 is too small. It is not readable for audience.

Round 2

Reviewer 2 Report

The authors addressed the comments from reviewer.